# Adolescents with Neuropsychiatric Disorders during the COVID-19 Pandemic: Focus on Emotional Well-Being and Parental Stress

**DOI:** 10.3390/healthcare10122368

**Published:** 2022-11-25

**Authors:** Francesca Felicia Operto, Costanza Scaffidi Abbate, Francesco Tommaso Piscitelli, Miriam Olivieri, Luigi Rizzo, Gianpiero Sica, Angelo Labate, Michele Roccella, Marco Carotenuto, Grazia Maria Giovanna Pastorino

**Affiliations:** 1Child and Adolescence Neuropsychiatry Unit, Department of Medicine, Surgery and Dentistry, University of Salerno, 84081 Salerno, Italy; 2Department of Psychology, University of Palermo, 90144 Palermo, Italy; 3Azienda Sanitaria Locale Salerno, via Nizza 146, 84124 Salerno, Italy; 4Neurophysiopatology and Movement Disorders Clinic, University of Messina, 98122 Messina, Italy; 5Department of Psychology, Educational and Science and Human Movement, University of Palermo, 90128 Palermo, Italy; 6Child and Adolescent Neuropsychiatry Clinic, Department of Mental and Physical Health and Preventive Medicine, University of Campania “Luigi Vanvitelli”, 80131 Naples, Italy

**Keywords:** COVID-19, neuropsychiatric disorders, adolescents, emotional behavioral symptoms, parental stress, CBCL, PSI

## Abstract

(1) Introduction: The aim of our research was to explore emotional/behavioral changes in adolescents with neuropsychiatric conditions during the COVID-19 pandemic, and parental stress levels through a standardized assessment, comparing the data collected before and during the first months of lockdown. Moreover, an additional goal was to detect a possible relationship between emotional/behavioural symptoms of adolescents and the stress levels of their parents. (2) Methods: We enrolled 178 Italian adolescents aged between 12–18 that were referred to the Child Neuropsychiatry Unit of the University Hospital of Salerno with different neuropsychiatric diagnoses. Two standardized questionnaires were provided to all parents for the assessment of parental stress (PSI-Parenting Stress Index-Short Form) and the emotional/behavioral problems of their children (Child Behaviour Check List). The data collected from questionnaires administered during the six months preceding the pandemic, as is our usual clinical practice, were compared to those recorded during the pandemic. (3) Results: The statistical comparison of PSI and CBCL scores before/during the pandemic showed a statistically significant increase in all subscales in the total sample. The correlation analysis highlighted a significant positive relationship between Parental Stress and Internalizing/Externalizing symptoms of adolescent patients. Age and gender did not significantly affect CBCL and PSI scores, while the type of diagnosis could affect behavioral symptoms and parental stress. (4) Conclusions: our study suggests that the lockdown and the containment measures adopted during the COVID-19 pandemic could have aggravated the emotional/behavioral symptoms of adolescents with neuropsychiatric disorders and the stress of their parents. Further studies should be conducted in order to monitor the evolution of these aspects over time.

## 1. Introduction

Since the World Health Organization (WHO) declared the global pandemic status on 11 March 2020 [1] and many countries established national quarantine, there have been drastic changes in people’s lifestyles, perception of illness and social interaction [2].

The restrictions that were progressively adopted to contain the propagation of the virus in Italy, as in the rest of the world, caused negative effects on the psychological well-being of the general population [3,4,5,6]. 

Stressors due to the pandemic and quarantine-related issues have shown a negative influence on the mental well-being of both adults and young people [7,8].Children and adolescents experienced an unexpected and abrupt interruption in their everyday lives, including a prolonged stay in the home, altered relationship dynamics with their caregivers and family members, more infrequent interactions with peers, changed nutritional and lifestyle behaviors and other changes at the household level [9,10]. 

Children and adolescents appear to be more sensitive to the effects of COVID-19 than adults in developing internalizing and externalizing symptoms such as anxiety, depression, irritability and sleep problems [11,12,13,14,15]. 

The systematic review by Alamolhoda and colleagues [16] revealed an overall negative impact on the well-being of teenagers. The authors reported that the most frequent problems were increased levels of stress, anxiety symptoms, sleep difficulties, mood disorders and post-traumatic stress disorder. 

Petruzzelli and colleagues [17] highlighted a significant increase in psychiatric counseling in the Emergency Room in 2020–2022 compared to 2019, particularly for acute psychopathological symptoms in adolescent patients. 

A study by Taylor et al. [18] found a significant increase in hair cortisol concentration in adolescents from pre- to post-lockdown of March 2020, which is robustly connected with neural dysfunction and consequent altered psychological well-being in the form of anxious and depressive symptoms.

In addition, the COVID-19 quarantine has been shown to impact the emotional well-being of young people with pre-existing neurological and psychiatric disabilities and their families, including increased levels of stress, anxiety and depression [19,20]. A piece of Italian research in 2021 conducted on a sample of patients with different neuropsychiatric disorders outlined a significant worsening in emotional/behavioral symptoms [21]. Another Italian study published in 2022, concerning children and adolescents with neuropsychiatric disorders, showed in the entire sample an aggravation as in internalizing as in externalizing symptoms during the lockdown period [22]. 

In the systematic review by Dal Pai et al. [23] the authors underlined that children and adolescents with Autism Spectrum Disorder (ASD) experienced several pandemic-related issues that impacted their psychological well-being, behavior and sleep patterns. Another study that examined stress and anxiety levels in youth with and without ASD and their parents during the first three months of the pandemic, revealed that ASD families experienced more anxiety and stress [24].

The study by Theis et al. [25] on children with Intellectual Disabilities (ID) showed negative consequences on mental health, mood, behavior, social interaction and learning in most of the analyzed patients. Zhang et al. [26] underlined an aggravation of inattention and hyperactivity symptoms during the COVID-19 pandemic in children affected by Attention-Deficit/Hyperactivity Disorder (ADHD), in addition to stress and mood alterations. The effects of the pandemic seem more severe in that part of the ADHD population which mostly uses the internet and electronic devices [27].A negative impact of the pandemic of emotional well-being was registered also in the pediatric population with epilepsy [28,29]. 

Finally, recent studies conducted on adolescents with [30,31,32,33,34,35,36] and without [37,38] neuropsychiatric disorders suggested a relationship between the emotional/behavioral symptoms of children and parental stress. Significant data were collected by Alhzumi [39] about children with ASD. Parental stress and emotional well-being of parents of children with ASD in Saudi Arabia had been unfavorably impacted by the COVID-19 pandemic.

Bitsika et al. [40] reported a raise in the risk of depression, anxiety and brooding thought in parents of children with Autism Spectrum Disorders; similarly, Willner et al. [41] noted heightened anxious and depressive traits and a sense of constraint in caregivers of children and adults with Intellectual Disabilities. 

The review by Rohwerderat al. [42] highlights both the scarcity of literature and the importance of conducting research specifically on adolescents with disabilities in humanitarian emergencies, since the COVID-19 pandemic had adverse effects on adolescents with disabilities across health, education, livelihoods, social protection and community participation domains from those of non-disabled adolescents.

Furthermore, Shorey et al. [43] revealed in their literature review a lack of evidence-based studies and articles on children with other neurodevelopmental disorders apart from ASD and ADHD.

Based on this evidence, the main aim of our research is to describe the psychological, emotional and behavioral changes in Italian adolescents with different neuropsychiatric conditions before and during the first national lockdown and the effect on parental stress. 

## 2. Materials and Methods

### 2.1. Participants

Our study was conducted at the Child and Adolescent Neuropsychiatric Unit of the San Giovanni di Dio e Ruggid’Aragona University Hospital in Salerno (Italy), a Complex Operative Unit for the diagnosis and pharmacological treatment of children and adolescents with neurological and psychiatric disorders (0–18 years).

We enrolled all patients aged 12–18 who were referred to our Neuropsychiatry Unit in the months preceding the pandemic (September 2019–January 2020—Time 0) and whose parents had completed two standardized questionnaires for the assessment of emotional/behavioral symptoms (Child BehaviorCheckList6–18years, CBCL) and Parenting Stress Index–Short Form (PSI), as in our usual neuropsychological evaluations (Figure 1).

All the participants had a previous neuropsychiatric diagnosis, received in our unit between 2018–2019 by a multidisciplinary team (child neuropsychiatrists, speech therapists and child psychologists), based on DSM-5 criteria and supported by clinical observations and standardized neuropsychological tests.

The parents of all recruited patients were remotely contacted during the first lockdown (March–May 2020—Time 1) and were asked to repeat the CBCL and PSI questionnaires. A detailed explanation about the purpose and the procedures of the study was provided to all parents who sent their informed consent in writing form by e-mail afterwards. The only exclusion criterion was the poor compliance of the parents. 

The parents who decided to participate in our study completed the questionnaires remotely, by different devices such as a smartphone or personal computer. Alternatively, the parents could fill out the questionnaires by e-mail or through an online program created for that specific purpose.

A team composed of six child psychologists and two child neuropsychiatrists analyzed the data collected during the lockdown. Subsequently, the data collected during the lockdown were compared with those collected before the pandemic.

In our analysis, we considered gender, age, diagnosis, comorbidities, drug therapy, age and level of education of the parents (years of education).

The study was approved by the Campania South Ethics Committee (protocol 61902) and followed the guidelines of good clinical practice.

### 2.2. Child Behavior Check List

The Child Behavior Checklist (CBCL) [44] is a standardized questionnaire that has been proven to be an effective way of assessing emotional/behavioral symptoms in children and adolescents aged between 6–18. The tool includes 113 statements and there are three possible options for each question: 2 = true or very true; 1 = sometimes true; 0 = false. The raw scores were converted into t-scores based on gender and age. The t-scores were distributed among 6 DSM-oriented subscales and 8 empirical subscales that can lead to obtain 3 main scales: Externalizing, Internalizing and Total Problems. We can interpret the t-score of all the subscales as follow: ≤64 = in the range of the norm; 65–69 = borderline; ≥70 pathological. Moreover, we can interpret the t-scores of the main scales based on the following interval: ≤59 = in the range of the norm; 60–64 = borderline; ≥65 pathological. Alpha reliability coefficient shave been noted to be ranged from 0.72–0.97. In addition, the test–retest reliability score was from 0.82–0.94.

### 2.3. ParentingStress Index

The Parenting Stress Index Short Form (PSI-SF) [45] is a standardized survey comprehending 36 questions with 5 possible answers for each. A score from 1 to 5 is assigned to each answer (1 = strongly disagree – 5 = strongly agree). Four scales emerged from this questionnaire: Parental Distress (PD), Parent–child Dysfunctional Interaction (P-CDI), Difficult Child (DC) and Total Stress (TS).The raw score is converted to t-scores based on age; t-score ≥85 is considered clinically significant. The reliability coefficients were ≥0.96, showing a high internal consistency. Test–retest reliability score was 0.65–0.96.

### 2.4. Statistical Analysis

We expressed all the neuropsychological scores as mean ± standard deviation (SD). We have preliminarily carried out an evaluation of the data distribution through the Kolmogorov–Smirnov normality test and we decided to employ non-parametric methods for our analysis, as many data did not meet the criteria of a normal distribution. For the comparison between mean scores before and during the pandemic, we used Wilcoxon signed-rank test. The possible relationship among different data was explored through the Spearman correlation test. We calculated the effect size using the two indicators η^2^ and r, interpreting them as follows: a value ≤ 0.3 represents a small effect size, a value between 0.4–0.7 represents a medium effect size, a value ≥ 0.8 represents a large effect size [46].Multivariate linear regression analysis and posthoc analysis were employed to evaluate the effect of some variables such as gender, age and type of diagnosis on CBCL and PSI scores. We considered statistically significant a *p*-value < 0.05.Statistical Package for Social Science software, version 23.0 (IBM Corp, 2015) was used for our statistical analysis.

## 3. Results

### 3.1. Sample Characteristics

We recruited 183 families but five (2.7%) decided to not participate in our study. The total sample included 178 Caucasian adolescents and their families, aged between 12–18 years (mean age = 15.34 ± 2.17; male = 108, 61%) with the following neuropsychiatric diagnosis: epilepsy (n = 66; 37%), autism spectrum disorder (n = 26; 15%), specific learning disorders (n = 26; 15%), anxiety disorders (n = 15; 8%), intellectual disability (n = 13; 7%), attention-deficit/hyperactivity disorder (n = 11; 6%), behavioral disorders (n = 11; 6%), mood disorders (n = 10; 6%). The main socio-demographic and clinical characteristics of all the participants were summarized in Table 1 and Table 2.

### 3.2. Mean Score Comparison of CBCL before/during the First Lockdown

The statistical comparison of the mean CBCL scores showed a significant increase (*p* < 0.05) at Time 1in all the scales (Anxiety/Depression, Withdrawal/Depression, Somatic complaints, Socialization, Thought problems, Attention problems, Rule-breaking behavior, Aggressive behavior, Affective problems, Anxiety problems, Somatic Problems, ADHD, Oppositional-defiant problems, Conduct problems, Internalizing problems, Externalizing problems, Total problems) (Table 3).

The effect size ranges from small to medium, according to the different scales, with a lower effect for the scale of Thought problems and a greater effect for the scales of Anxiety/Depression, Somatic complaints, Anxiety problems, Internalizing problems and Total problems (Table 3).

Analyzing the subsamples divided by diagnoses, we found that all the CBCL scales were significantly higher in adolescents with specific learning disorders, anxiety, behavioral disorder and epilepsy (*p* < 0.05).

In adolescents with Autism, we found a significant increase in all the CBCL scales (*p* < 0.05) except for pervasive problems (Z = −1.450, *p* = 0.147) and attention problems (Z = −1.851, *p* = 0.064). In the adolescents with intellectual disabilities all the CBCL scales were significantly higher at Time 1 except Thought problems (Z = −1.409, *p* = 0.159) and ADHD (Z = −1.691, *p* = 0.091). We did not find significant changes in the CBCL scales in the ADHD subsample, excepting for the Somatic complaints scale (Z = −2134, *p* = 0.033). In the adolescent with mood disorder all the CBCL subscales were significantly higher (*p* < 0.05) except for Thought problems (Z = −0.250, *p* = 0.803).

### 3.3. Mean Score Comparison of PSI before/during the First Lockdown

We found a statistically significant increase at Time 1 compared to Time 0 in all the mean scores of PSI (Parental Distress, Parent–child Difficult Interaction, Difficult Child, Total Stress) (Table 3). The effect size ranges from small to medium, with a lower effect for the Difficult Child scale and a greater effect for the Total stress scale (Table 3).

Analyzing the different subsamples divided by diagnosis emerged that all the PSI scales were significantly higher in patients with autism, ADHD, specific learning disorder, anxiety and behavioral disorder (*p* < 0.05). 

All the PSI scales were significantly higher (*p* < 0.05) in the epilepsy subsample except for Parent–child Dysfunctional Interaction scale (Z = −1.364, *p* = 0.173).All of the mean scores of PSI scales were significantly higher (*p* < 0.05)in the intellectual disability subpopulation except for Difficult Child (Z = −1.436, *p* = 0.151). We did not find significant changes in the PSI scales in the mood disorder group (*p* > 0.05).

### 3.4. Correlation Analysis between PSI and CBCL Scales

The correlation analysis showed a significant positive relationship between all the subscales of PSI (Parental Distress, Parent–child Difficult Interaction, Difficult Child and Total Stress) and the main scales of CBCL (Total problems, Externalizing problems, Internalizing problems). The strength was medium for all of the relationships analyzed. All the results of the correlation analysis are summarized in Table 4.

### 3.5. Multivariate Linear Regression Analysis at Time 1

We performed multivariate linear regression analysis to investigate the effect of sex, age and diagnosis variables on CBCL and PSI scores at Time 1. We found that gender and age did not statistically significantly affect CBCL scores and PSI (Table 5), while the type of diagnosis has a statistically significant impact (Table 5).

The posthoc analysis of CBCL scores showed statistically significant differences, as follows: the Total problems scale was significantly lower in patients with epilepsy than in patients with autism spectrum disorder (*p* = 0.017) and behavioral disorders (*p* = 0.005); the Externalizing problems scale was significantly lower in patients with epilepsy than in patients with autism spectrum disorder (*p* = 0.008), anxiety disorder (*p* = 0.008), mood disorder (*p* = 0.013) and behavioral disorder (*p* < 0.001); the Internalizing problems scale was lower in patients with epilepsy than in patients with behavioral disorder, but did not reach the statistical significance (*p* = 0.068). 

The posthoc analysis of PSI scores showed the following results: The Parental Distress scale (PD) was significantly lower in patients with epilepsy than in those with autism spectrum disorder (*p* = 0.003), anxiety disorder (*p* = 0.001), ADHD (*p* = 0.044), behavioral disorder(*p* = 0.017), and was significantly higher in patients with an anxiety disorder than in patients with a specific learning disorder (0.043). The Parent–child Dysfunctional Interaction scale (P-CDI) was significantly lower in patients with epilepsy than in those with autism spectrum disorder (*p* = 0.046) and anxiety disorder (*p* = 0.037). The Difficult Child (DC) was significantly lower in patients with epilepsy than in those with autism spectrum disorder (*p* = 0.021). The Total Stress scale (TS) was significantly lower in patients with epilepsy than in those with autism spectrum disorder (*p* = 0.005), anxiety disorder (*p* = 0.004) and behavioral disorder (*p* = 0.023).

## 4. Discussion

The aim of our research was to assess the psychological impact of the COVID-19 pandemic on an Italian adolescent population with neurological and psychiatric disorders and parental stress, comparing the pre-pandemic period with the first month of lockdown. 

During the COVID-19 lockdown, children and adolescents suffered from isolation, confinement, boredom and worries about their psychophysical health [47,48,49,50,51,52], with increased concerns in parents about their children’s well-being [53,54]. Families with children with special needs presented the most serious consequences of the lockdown [55]. It is important to underline the close relationship between the parents’ and children’s emotional status, with significant repercussions on the entire families’ psychological well-being. The experience of specific worries about their children’s mental well-being increases the risk of depression, brooding and anxiety in their parents’ well-being [56]. On the other hand, increased parental anxiety and worries were related to emotional dysregulation and internalizing and externalizing problems in their children during the pandemic [57]. 

Our sample included 178 Italian adolescents and their families (Table 1 and Table 2), aged between 12–18 years with the following neuropsychiatric diagnosis: epilepsy (n = 66), autism spectrum disorder (n = 26), specific learning disorders (n = 26), anxiety disorders (n = 15), intellectual disability (n = 13), attention-deficit/hyperactivity disorder (n = 11), behavioral disorders (n = 11), mood disorders (n = 10). All the parents completed two standardized questionnaires for the assessment of emotional/behavioral symptoms of the adolescents (CBCL) and of parental stress (PSI) before/during the COVID-19 pandemic.

Overall, our study showed a significant increase in both internalizing and externalizing symptoms in the whole sample during the lockdown, as reported by the parents. The statistical comparison before/during the lockdown showed a significant worsening in all scales of the CBCL, with a greater effect for the scales of Anxiety/Depression, Somatic complaints, Anxiety problems, Internalizing problems and Total problems. These findings suggest that internalizing problems, such as anxiety and somatic symptoms, had a greater impact in our sample.

Our results agree with a literature review by Guessoum et al., (2020) [13] that confirmed an increase in the symptomatology in adolescents with a psychiatric disorder, especially mood disorders, ADHD and autism spectrum disorder. Doyle and colleagues [58] detected a worsening in psychiatric symptoms of 171 youth outpatients (mean age aged 10.6 ± 3.1)during the school year following the first COVID-19 lockdown, including anxiety, oppositional behavior and hyperactivity/impulsivity. Conti et al. [21] registered a worsening of emotional and behavioral traits in 141 patients under the age of 18 (mean age = 10.6 ± 3.1 years) with neuropsychiatric disorders during the March 2020 lockdown. In addition, a tendency for a better reaction to the lockdown in psychiatric patients than the neurological disorders group has been suggested. Conversely, Raffagnato and colleagues [31] did not observe major changes during the lockdown compared to the previous period in a sample of children and adolescents with mental health disorders (n = 56; mean age = 13.4 ± 2.77 years). In particular, the authors concluded that the group of patients who previously suffered from internalizing disorders overall showed a good adaptation to the pandemic context, while a greater psychological discomfort was detected in patients with behavioral problems attributable to neurodevelopmental and conduct disorders. Finally, our results are at odds with those reported by De Giacomo et al. [35], which did not detect a worsening of the internalizing and externalizing CBCL symptoms in 71 children with neuropsychiatric disorders before and during the lockdown (average age 9.01 ± 3.67). This difference may be due to the different ages of our sample, which only considered adolescents.

Analyzing the subsamples divided by diagnoses, we found that the adolescents with anxiety disorder, mood disorder, specific learning disorder, behavioral disorder and epilepsy showed a global worsening in all the CBCL scales. In agreement with our data, previous literature studies had shown a worsening of psychological well-being during the lockdown in patients with epilepsy and learning disabilities [28,29,59,60,61] probably due to difficulties with distance learning; in contrast with our results, a larger sample study found that both depressive and anxiety symptoms were lower for adolescents with mood disorders during the pandemic compared to before [62].

In the subsample with Autism, we found a significant worsening of all the CBCL scales, except for pervasive problems and attention problems. In the adolescents with intellectual disabilities all the CBCL scales were significantly higher except for Thought problems and ADHD scales. Several previous studies confirmed a worsening in young people with Autism Spectrum Disorder and Intellectual Disabilities during the COVID-19 pandemic, mainly represented by an increase in anxiety symptoms, irritability hyperactivity and behavioral problems [23,24,25,63], probably due to changes in their daily routines and access to therapies. Other studies, on the other hand, suggested a reduction in emotional behavioral manifestations in adolescents with autism due to lower social pressure [64].

In our study, surprisingly, we did not find significant changes in the CBCL scales in the ADHD subsample, except for the Somatic complaints scale. These results are in contrast with previous surveys [26,27]. Behrmann et al. [65] observed in adolescent patients diagnosed with ADHD, who experience depression, anxiety, loneliness, boredom and emotional distress. We can speculate that the lockdown hit differently children and adolescent’s psychological well-being, considering that only adolescent subjects have been observed in our analysis.

Analyzing the PSI scores we found a significant increase in parental stress during the lockdown in all the areas analyzed(Parental Distress, Parent–child Difficult Interaction, Difficult Child, Total Stress), with a lower effect for the Difficult Child scale and a greater effect for the Total stress scale. The feeling of having a complicated relationship with children was strongly heightened during the lockdown period, highlighting a greater perception of parental-role-related stress and total stress.

Analyzing the different subsamples that emerged, all the PSI scales were significantly higher in patients with autism, ADHD, specific learning disorder, anxiety and behavioral disorder. All the PSI scales were significantly higher in the epilepsy group except for Parent–child Dysfunctional Interaction scale. All the mean scores of PSI scales were significantly higher in the intellectual disability subpopulation, except for Difficult Child. The results of our study are in agreement with previous literature studies, which demonstrated increased parental stress in families with children and adolescents with neurological or psychiatric problems [66]. The study of De Giacomo et al. [35] highlights an increase in parental stress and a more difficult parent–child interaction in the period of lockdown due to the pandemic in a sample of children with neuropsychiatric conditions. Contrary to what we expected, we did not find significant changes in the PSI scales in parents of adolescents with mood disorders. We can hypothesize that internalizing symptoms, such as anxiety and mood disorders, could be hard to acknowledge and decipher; for this reason, these pathological states could worsen without being noticed. 

The correlation analysis showed a significant relationship between parental stress and the internalizing/externalizing symptoms of adolescent patients. In this regard, it was shown that the resilience of the entire family has a mutual influence on mental well-being and the ability to successfully cope with changes, both in parents and children, resulting in stress and depression of children related to those of parents [37]. In agreement with our results, Costa et al. [67] suggest that this relationship could be due to parents’ dissatisfied expectations of their children or else to a non-reinforcing parent–child interaction. These data are supported also by the study by Sesso et al. [30] that, in a cross-sectional study on pediatric patients with neuropsychiatric conditions during Lockdown, found significant positive associations between CBCL Internalizing problems of and all PSI subscales, and between CBCL Externalizing problems and Difficult Child subscales.

The multivariate regression analysis in our sample highlighted that age and sex did not affect the emotional/behavioral symptoms and parental stress during the lockdown. On the contrary, the type of diagnosis could significantly affect both. More in detail, patients with epilepsy showed less emotional/behavioral problems, especially with regard to externalizing symptoms than adolescents with autism spectrum disorder, anxiety disorder, mood disorder and behavioral disorder. Finally, parental stress levels were lower in the epilepsy group than in the groups diagnosed with autism spectrum disorder, anxiety disorder, specific learning disorder, ADHD, and behavioral disorder. This finding suggests that emotional–behavioral symptoms may be more present in pediatric patients with psychiatric disorders than in patients with epilepsy during the COVID-19 lockdown. Similarly, although high levels of parental stress are documented in the parents of patients with different neuropsychiatric conditions [68,69,70,71], parents of adolescents with psychiatric disorders may experience higher levels of stress than those of adolescents with epilepsy.

The strength of the study was the use of standardized quantitative questionnaires and the focus on a specific population of patients (adolescents with neuropsychiatric disorders). The main limitation of our study is the small sample size, especially in some subgroups, which can reduce the statistical power. Other weaknesses of the study are the lack of a control group and the homogeneous geographical provenience of the participants. In future studies, we aim to analyze the emotional/behavioral symptoms of adolescents with neuropsychiatric disorders and parental stress over time (during and post the COVID-19 pandemic) and compare them with a control group. We aim to consider, in our future analysis, other socio-demographic and clinical variables than can affect psychological well-being during the COVID-19 pandemic.

## 5. Conclusions

Our research showed a significant worsening of internalizing and externalizing symptoms in adolescents with neuropsychiatric conditions during the first COVID-19 lockdown, which could suggest that confinement, social isolation, lifestyle changes and reduced access to therapy could impact the global psychological well-being in this population. Our study showed also an increase in parental stress, that was related to both internalizing and externalizing symptoms of the adolescent patients. We can suppose that the malaise in individual family members, in addition, to reiterate intra-familiar interactions, could have set up a vicious circle of parental stress and adolescents’ behavioral and emotional symptoms.

## Figures and Tables

**Figure 1 healthcare-10-02368-f001:**
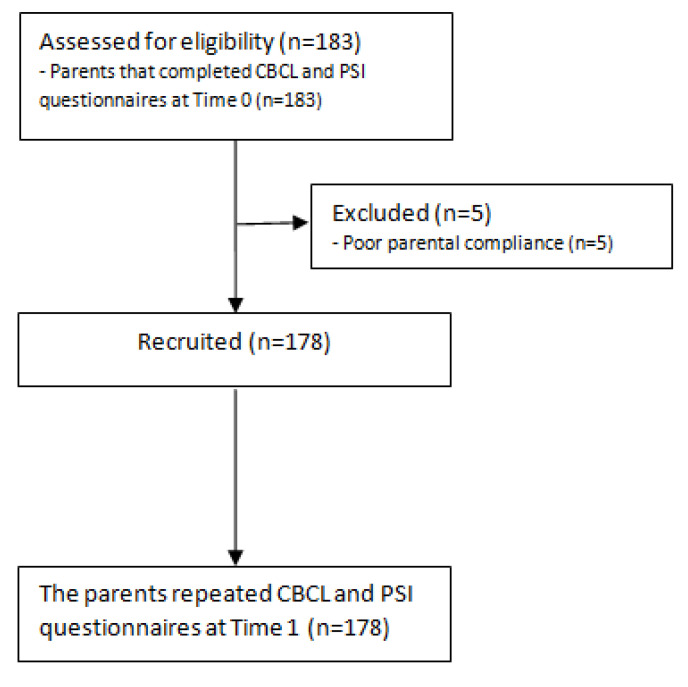
Flow-chart of the research procedure. CBCL, Child Behavior Check List; PSI, Parenting Stress Index; Time 0 = September 2019–January 2020; Time 1 = March 2020–May 2020.

**Table 1 healthcare-10-02368-t001:** Main socio-demographic feature of the total sample. SD = standard deviation. * calculated in years of education.

Participants n = 178	
child age (mean ± SD)	15.34 ± 2.17
sex	
male	108 (61%)
female	70 (39%)
father age (mean ± SD)	46.28 ± 6.39
mother age (mean ± SD)	43.63 ± 5.70
maternal education level (mean ± SD) *	13.98 ± 4.16
paternal education level (mean ± SD) *	14.00 ± 3.97

**Table 2 healthcare-10-02368-t002:** Socio-demographic and clinical features of the subsample and total sample.

Principal Diagnosis	Age (Years)	Sex	Father Age (Years)	Mother Age (Years)	Neuro-Psychiatric Comorbidities	Other Clinical Conditions	Drug Therapy
epilepsy n = 66	15.99 ± 1.96	male = 41 (62%)	48.05±6.02	45.18±5.52	20 (30%)	14 (21%)	60 (91%)
autism spectrum disorder n = 26	15.35 ± 2.61	male = 18 (69%)	49.68±6.94	47.24±6.30	17 (65%)	10 (38%)	9 (345%)
specific learning disorders n = 26	13.96 ± 1.91	male = 15 (58%)	41.85±4.71	39.38±3.71	6 (23%)	5 (19%)	0 (0%)
anxiety disorders n = 15	15.17 ± 2.22	male = 6 (40%)	45.40±7.08	43.80±4.52	8 (53%)	2 (13%)	12 (80%)
intellectual disability n = 13	15.92 ± 1.85	male = 10 (77%)	45.31±4.75	42.31±5.53	8 (61%)	4 (31%)	3 (23%)
attention-deficit/hyperactivity disorder n = 11	15.09 ± 4.38	male = 7 (64%)	45.09±11.91	42.09±11.10	3 (27%)	3 (27%)	9 (82%)
behavioral disorders n = 11	15.09 ± 3.30	male = 7 (64%)	44.36±9.36	41.73±9.00	3 (27%)	2 (18%)	5 (45%)
mood disorders n = 10	14.70 ± 2.07	male = 4 (40%)	44.00±7.94	40.70±5.19	7 (70%)	2 (20%)	10 (100%)
Total sample n = 178	15.34 ± 2.17	male = 108 (61%)	46.28±6.39	43.63±5.70	72 (40%)	42 (24%)	108 (61%)

**Table 3 healthcare-10-02368-t003:** Statistical comparison between mean scores of the Child Behavior Check List and the Parental Stress Index at time 0 (before the pandemic) and at Time 1 (March-May 2020). *p*-value < 0.05 are in bold. N = 178.

Standardized Neuropsychologicaltest	Time 0 (Mean ± SD)	Time 1 (Mean ± SD)	Statistic (Wilcoxon Test)	*p*-Value	η^2^ = z^2^/N − 1	r = z/√(N × 2)
Parental Stress Index (PSI)						
Parental Distress (PD)	59.86 ± 28.64	72.67 ± 27.80	z = −7.918	***p* < 0.001**	0.354	0.420
Parent–child Difficult Interaction (P-CDI)	63.82 ± 23.78	77.58 ± 23.44	z = −7.703	***p* < 0.001**	0.335	0.409
Difficult Child (DC)	65.10 ± 26.35	77.48 ± 24.99	z = −7.242	***p* < 0.001**	0.296	0.385
Total Stress (TS)	63.68 ± 25.09	77.19 ± 25.22	z = −8.487	***p* < 0.001**	0.406	0.451
Child Behavior Check List (CBCL) 6–18 years				***p* < 0.001**		
Anxiety/Depression	59.52 ± 7.66	69.49 ± 13.24	z = −9.113	***p* < 0.001**	0.469	0.484
Withdrawal/Depression	61.38 ± 8.60	69.92 ± 12.99	z = −8.391	***p* < 0.001**	0.397	0.446
Somatic complaints	58.35 ± 7.86	69.60 ± 13.65	z = −8.642	***p* < 0.001**	0.421	0.459
Socialization	62.59 ± 9.05	70.66 ± 13.04	z = −7.974	***p* < 0.001**	0.359	0.423
Thought problems	61.67 ± 10.57	66.08 ± 11.90	z= −5.342	***p* < 0.001**	0.161	0.283
Attention problems	62.01 ± 8.85	68.88 ± 11.78	z = −8.255	***p* < 0.001**	0.385	0.438
Rule-breaking behavior	58.44 ± 7.43	65.61 ± 12.59	z = −8.267	***p* < 0.001**	0.386	0.439
Aggressive behavior	61.28 ± 10.99	67.35 ± 13.26	z = −7.695	***p* < 0.001**	0.334	0.409
Affective problems	61.93 ± 7.85	69.09 ± 11.73	z = −8.501	***p* < 0.001**	0.408	0.451
Anxiety problems	61.96 ± 7.70	69.95 ± 11.70	z = −8.796	***p* < 0.001**	0.437	0.467
Somatic Problems	56.90 ± 8.06	65.08 ± 11.39	z = −8.058	***p* < 0.001**	0.366	0.428
ADHD	60.46 ± 7.66	66.62 ± 10.17	z= −8.036	***p* < 0.001**	0.364	0.427
Oppositional-defiant problems	58.33 ± 7.66	64.57 ± 11.26	z= −8.006	***p* < 0.001**	0.362	0.425
Conduct problems	57.28 ± 7.13	63.29 ± 11.08	z = −8.110	***p* < 0.001**	0.371	0.431
Internalizing problems	59.66 ± 10.24	68.89 ± 13.35	z = −9.017	***p* < 0.001**	0.459	0.479
Externalizing problems	57.65 ± 10.28	65.88 ± 14.22	z = −8.198	***p* < 0.001**	0.379	0.435
Total Problem	60.17 ± 9.87	69.19 ± 13.93	z = −9.159	***p* < 0.001**	0.473	0.486

**Table 4 healthcare-10-02368-t004:** Spearman correlation analysis between Child Behavior Checklist and Parental Stress Index subscales. CBCL = Child Behavior Checklist. *p*-value < 0.05 are in bold.

			CBCL Total Problems	CBCL Externalizing Problems	CBCL Internalizing Problems
Parental stress	Parental Distress	r	0.454	0.407	0.466
		*p*-value	**<0.001**	**<0.001**	**<0.001**
	Parent–child Difficult Interaction	r	0.490	0.401	0.499
		*p*-value	**<0.001**	**<0.001**	**<0.001**
	Difficult Child	r	0.504	0.424	0.495
		*p*-value	**<0.001**	**<0.001**	**<0.001**
	Total Stress	r	0.546	0.466	0.556
		*p*-value	**<0.001**	**<0.001**	**<0.001**

**Table 5 healthcare-10-02368-t005:** Multivariate linear regression analysis between at Time 1. *p* value < 0.05 are in bold.

		Sex	Age	Diagnosis
Child Behavior Check List (CBCL)	Total Problems	*p* = 0.261	*p* = 0.849	***p* < 0.001**
	Externalizing Problems	*p* = 0.657	*p* = 0.879	***p* < 0.001**
	Internalizing	*p* = 0.294	*p* = 0.562	***p* < 0.001**
Parental Stress Index (PSI)	Parental Distress	*p* = 0.388	*p* = 0.320	***p* < 0.001**
	Parent–child Dysfunctional Interaction	*p* = 0.644	*p* = 0.669	***p* = 0.001**
	Difficult Child	*p* = 0.822	*p* = 0.710	***p* = 0.005**
	Total Stress	*p* = 0.613	*p* = 0.432	***p* < 0.001**

## Data Availability

The data presented in this study are available on request from the corresponding author.

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
