# Peer review of "Adolescents with Neuropsychiatric Disorders during the COVID-19 Pandemic: Focus on Emotional Well-Being and Parental Stress"

_healthcare, 2022, doi:10.3390/healthcare10122368_

Round 1

Reviewer 1 Report

Operto et al. investigated the emotional well-being and parental stress in Italian adolescents with neuropsychiatric conditions.  They compared timepoints of pre-pandemic and soon into lockdown using behavioral questionnaires.  While the topic is of interest to Healthcare readers, in its current form, this paper fails to make an original contribution to the literature.  I have concerns about the the lack of thoroughness and clarity in summarizing and critically evaluating existing literature on the mental health impact of the pandemic in adolescents.  I also have concerns about the research design, presentation of results, and conclusions the authors made that are not supported by their statistical methods or results.  I appreciate the authors efforts and hope that this feedback is beneficial.

General:

-This paper would benefit from editorial support to enhance use of the English language and academic tone.  There are also several typos throughout that should be edited (e.g., Title on page 1).

Introduction:

-In general, this section would be improved by summarizing and critically evaluating the current literature on emotional well being in adolescents during COVID.  For example, what is missing in the current research that is addressed by the authors' study?  Can the authors clearly summarize existing literature so that the reader can clearly understand where the findings agree or disagree?  

-The authors may consider focusing their review of existing literature to provide a clear rationale for why they chose to study this population (adolescents with neuropsychiatric disorders), why they chose the time periods they did, and why they are interested in the internalizing/externalizing symptoms they chose to study.  In its current form, it is difficult to ascertain support for the authors' aims.

-The authors appear to have misinterpreted some of the existing literature they cited in the introduction.  For example, they state that "Besides, the qurantine had important consequences on paediatric population with pre-exisitng disorder.  A global worsening of behavioural symptoms..." (line 79-80 on pg 2).  However, the article they cited to support this statement consisted of research on a providence in China, not a global examination.  It is difficult to determine when they are summarizing research on pandemic-specific changes in the mental health of adolescents versus general changes that could be attributed to general development. 

-Some of the authors' summaries are not cited.  (For example, lines 66-72 on page 2, is this information all from one study?)

-This section would be improved by more clearly differentiating between summarizing/critiquing existing research on adolescent psychological well being and parent stress. 

-Lines 119-126 on pg 3 appear to be better suited for the methods section (discussing the questionnaires and population)

Methods:

-In general, more clarity is needed throughout this section to better understand the authors' research and statistical design. 

-Please provide further information on the method used to recruit participants.  How were they approached for the study?  How many were approached versus how many participated? 

-The authors shared that participants were patients in a hospital for neuropsychiatric clinic.  More information about this clinic is needed to help the reader understand the population.  Is it a diagnosis center?  A treatment center?  How are patients referred to the hospital?

-The authors refer to the PSI-SF and CBCL as "neuropsychological standardized questionnaires (line 137-138 on pg 3); however they are actually behavioral questionnaires that are used during neuropsychological evaluations.

-Please provide more specific information on the timepoints selected to complete questionnaires.  On line 143, pg. 3 the authors stated  they compared "the data registered before and during the pandemic".  It would be helpful to more clearly define these timepoints.

-The authors may wish to consult with a statistician for support regarding their statistical analyses.  It seems they made multiple group comparisons that might be better served with other statistical analyses (instead of Wilcoxen).

-It would be helpful to include additional demographics (e.g., race/ethnicity) to help the reader better understand the participants used.  Also, it helps the reader to determine generalizability of these results.

-Can the authors clarify what is meant by "calculated in age of school" on Table 1?  Is it years of education?

-Editing tables for format/typos would be helpful

Results:

-Again, consultation with a statistician may be helpful for presenting results.  There are a lot of subscales here and it is not presented clearly.  

-It seems there are missing results not presented in the text or in a table, but are frequently referred to in the discussion section.  Where are results on CBCL and PSI scales by diagnosis? 

-The authors should develop their presentation of correlation results more (lines 217-218, pg 6).  For example, the relationship is positive, but what does that mean in terms of their data?

Discussion:

-This section would benefit from significant revision.  Primarily, the authors should only present and discuss conclusions supported by their data.  For example, the authors stated that "New stressors emerged during the pandemic, arousing new fears and increasing previous pathological traits in both parents and children with neuropsychiatric conditions.  The malaise in individual family members, in addition to reiterate intra-familiar interactions, could have set up a vicious circle...." (lines 331-336 on pg 9).  However, they did not have data on new stressors, aspects of the lockdown that may have contributed to the increase in ratings they observed, family interactions, etc.--they only have data on emotional and behavioral ratings.  The authors also imply causation where they only studied correlation (e.g., lines 259-261 on pg 7).  

-Some paragraphs in this section seem to be simply reviewing other studies rather than discussing the implications of this study's results (e.g., lines 225-233 on pg 7).  It would be helpful to focus this section on study results, impact, etc. 

-On lines 254 -255 (pg 7), the authors noted a "comparable increase" in psycholgical symptoms in subgroups (Anxiety disorders, behavioral disorders, epilepsy, etc.).  However, they did not appear to make statistical comparisons between subgroups.

-Data discussed in this section does not appear in the results.  For example, the authors stated that "no negative alterations in CBCL subscales have been noticed in the group diagnosed with ADHD (lines 270-271, pg 8).  however, they did find a significant change between the timepoints they compared.  I could not find the results that showed CBCL/PSI ratings by subgroup.

Author Response

Operto et al. investigated the emotional well-being and parental stress in Italian adolescents with neuropsychiatric conditions.  They compared time points of pre-pandemic and soon into lockdown using behavioral questionnaires.  While the topic is of interest to Healthcare readers, in its current form, this paper fails to make an original contribution to the literature.  I have concerns about the the lack of thoroughness and clarity in summarizing and critically evaluating existing literature on the mental health impact of the pandemic in adolescents.  I also have concerns about the research design, presentation of results, and conclusions the authors made that are not supported by their statistical methods or results.  I appreciate the authors efforts and hope that this feedback is beneficial.

  • General:

This paper would benefit from editorial support to enhance use of the English language and academic tone.  There are also several typos throughout that should be edited (e.g., Title on page 1).

Authors ‘response: we thank the reviewer for the suggestion. The manuscript was revised by a native English speaker.

2) Introduction:

 -In general, this section would be improved by summarizing and critically evaluating the current literature on emotional well being in adolescents during COVID.  For example, what is missing in the current research that is addressed by the authors' study?  Can the authors clearly summarize existing literature so that the reader can clearly understand where the findings agree or disagree?  

Authors ‘response: we thank the reviewer for the comment. The Introduction section was improved, and we critically evaluated the current literature, as suggested. We also specified what was missing in the literature that was instead addressed in our work.

-The authors may consider focusing their review of existing literature to provide a clear rationale for why they chose to study this population (adolescents with neuropsychiatric disorders), why they chose the time periods they did, and why they are interested in the internalizing/externalizing symptoms they chose to study.  In its current form, it is difficult to ascertain support for the authors' aims.

Authors ‘response: we thank the reviewer for the comment. We specified what was missing in the literature that was instead addressed in our work in the Introduction section, as follows:" The review of Rohwerder at al. [42] highlights both the scarcity of literature and the importance of conducting research specifically with adolescents with disabilities in humanitarian emergencies since COVID-19 pandemic had adverse effects on adolescents with disabilities across health, education, livelihoods, social protection, and community participation domains from those of non-disabled adolescents. Furthermore, Shorey et al. [43] revealed in their literature review a lack of evidence-based studies and articles on children with other neurodevelopmental disorders apart from ASD and ADHD. Based on these evidence, the main aim of our research is to describe the psychological, emotional and behavioural changes in Italian adolescents with different neuropsychiatric conditions before and during the first national lockdown and the effect on the parental stress."

-The authors appear to have misinterpreted some of the existing literature they cited in the introduction.  For example, they state that "Besides, the qurantine had important consequences on paediatric population with pre-exisitng disorder.  A global worsening of behavioural symptoms..." (line 79-80 on pg 2).  However, the article they cited to support this statement consisted of research on a providence in China, not a global examination.  It is difficult to determine when they are summarizing research on pandemic-specific changes in the mental health of adolescents versus general changes that could be attributed to general development. 

Authors ‘response: we thank the reviewer for the comment. We deleted this sentence and we added other works that clearly described this topic.

-Some of the authors' summaries are not cited.  (For example, lines 66-72 on page 2, is this information all from one study?).

Authors ‘response: we thank the reviewer for the comment. We have completely revisited the Introduction through a critical review and summary of the literature; we added several new studies to describe current knowledge on the impact of covid-19 on the general population, adolescents, and people with neuropsychiatric disorders.

-This section would be improved by more clearly differentiating between summarizing/critiquing existing research on adolescent psychological well being and parent stress. 

Authors ‘response: we thank the reviewer for the comment. We have differentiated in two different paragraphs the data on the well-being of adolescents and those on parental stress

-Lines 119-126 on pg 3 appear to be better suited for the methods section (discussing the questionnaires and population),

Authors ‘response: we thank the reviewer for the comment. We have moved this paragraph to the methods section, as suggested

3) Methods:

-In general, more clarity is needed throughout this section to better understand the authors' research and statistical design. 

Authors ‘response: we thank the reviewer for the comment. As suggested, we have better specified the statistical design in the appropriate section.

-Please provide further information on the method used to recruit participants.  How were they approached for the study?  How many were approached versus how many participated? 

Authors ‘response: we thank the reviewer for the comment. We have better specified the recruitment method, and we have inserted a flowchart (figure 1).

 -The authors shared that participants were patients in a hospital for neuropsychiatric clinic.  More information about this clinic is needed to help the reader understand the population.  Is it a diagnosis center?  A treatment center?  How are patients referred to the hospital?

Authors ‘response: we thank the reviewer for the comment. We added this information in the methods section, as required.

-The authors refer to the PSI-SF and CBCL as "neuropsychological standardized questionnaires (line 137-138 on pg 3); however they are actually behavioral questionnaires that are used during neuropsychological evaluations.

Authors ‘response: we thank the reviewer for the comment, and we apologize for our mistake. We have corrected this point, as suggested.

-Please provide more specific information on the timepoints selected to complete questionnaires.  On line 143, pg. 3 the authors stated  they compared "the data registered before and during the pandemic".  It would be helpful to more clearly define these timepoints.

Authors ‘response: we thank the reviewer for the comment. We clearly specified the time points as required.

-The authors may wish to consult with a statistician for support regarding their statistical analyses.  It seems they made multiple group comparisons that might be better served with other statistical analyses (instead of Wilcoxen).

Authors ‘response: we thank the reviewer for the comment. We asked for the support of a statistician and, according to his indications, we added the effect size of the statistical comparisons made through the Wilcoxon test in order to give a dimension of the effect of our analyzes; we also added a multivariate linear regression analysis to analyze the influence of some factors on CBCL and PSI during the lockdown period.

-It would be helpful to include additional demographics (e.g., race/ethnicity) to help the reader better understand the participants used.  Also, it helps the reader to determine generalizability of these results.

Authors ‘response: we thank the reviewer for the comment. We added this information in the results section (all the participants are Caucasian)

-Can the authors clarify what is meant by "calculated in the age of school" in Table 1?  Is it years of education?

Authors ‘response: we thank the reviewer for the comment, and we apologize for our mistake. We meant "years of education," and we corrected the legend of the table.

-Editing tables for format/typos would be helpful.

Authors ‘response: we thank the reviewer for the comment. We corrected typos and we edited the tables.

4) Results:

-Again, consultation with a statistician may be helpful for presenting results.  There are a lot of subscales here and it is not presented clearly. 

 Authors ‘response: we thank the reviewer for the comment. We consulted a statistician, and we remodeled the presentation of the results.

-It seems there are missing results not presented in the text or in a table, but are frequently referred to in the discussion section.  Where are results on CBCL and PSI scales by diagnosis? 

 Authors ‘response: we thank the reviewer for the comment. We apologize for the mistake; we have completely reorganized the results and the discussion section, also in light of the new statistical results.

-The authors should develop their presentation of correlation results more (lines 217-218, pg 6).  For example, the relationship is positive, but what does that mean in terms of their data?

Authors ‘response: we thank the reviewer for the comment. We specified the mean of correlation data in the discussion sections.

5) Discussion:

-This section would benefit from significant revision.  Primarily, the authors should only present and discuss conclusions supported by their data.  For example, the authors stated that "New stressors emerged during the pandemic, arousing new fears and increasing previous pathological traits in both parents and children with neuropsychiatric conditions.  The malaise in individual family members, in addition to reiterate intra-familiar interactions, could have set up a vicious circle...." (lines 331-336 on pg 9).  However, they did not have data on new stressors, aspects of the lockdown that may have contributed to the increase in ratings they observed, family interactions, etc.--they only have data on emotional and behavioral ratings.  The authors also imply causation where they only studied correlation (e.g., lines 259-261 on pg 7).  

Authors' response: we thank the reviewer for the comment. We completely revisited this section, as required. The sentence "New stressors…" was deleted. The causation was replaced with correlation.

-Some paragraphs in this section seem to be simply reviewing other studies rather than discussing the implications of this study's results (e.g., lines 225-233 on pg 7).  It would be helpful to focus this section on study results, impact, etc. 

Authors' response: we thank the reviewer for the comment. We revisited the discussion section, and we tried to discuss the results of the other studies, highlighting the implications of our study.

-On lines 254 -255 (pg 7), the authors noted a "comparable increase" in psycholgical symptoms in subgroups (Anxiety disorders, behavioral disorders, epilepsy, etc.).  However, they did not appear to make statistical comparisons between subgroups.

Authors' response: we thank the reviewer for the comment. We agree with the reviewer, and since this paragraph was confusing, we have rephrased it.

-Data discussed in this section does not appear in the results.  For example, the authors stated that "no negative alterations in CBCL subscales have been noticed in the group diagnosed with ADHD (lines 270-271, pg 8).  however, they did find a significant change between the time points they compared.  I could not find the results that showed CBCL/PSI ratings by subgroup.

Authors' response: we thank the reviewer for the comment. We agree with the reviewer, and now we have ensured that all the results in the discussion section are also reported in the results section.

Reviewer 2 Report

The authors should ask for the help of a native English-speaking proofreader

because there are some linguistic mistakes that should be fixed. The title

needs further thought - shortened and more accurate. 

The Abstract in its sub-sections needs re-organization and it does not

adequately summarise the gist of the study.

Also, the writing style of the manuscript is not overall academic and formal.

The article is proposed to be supplemented with a flowchart illustrating the

research technique. A review of the literature is insufficient. It is critical to

include some recent work (2018–2020) in the literature review. A literature

review should be added in order to illustrate the central topic in a more

detailed way. Some further explanations and interpretations are required for

the results.

It is recommended to include a well-organized discussion of the

findings, strengths, and limitations of the present project with additional

explanation/details and a conclusion with future work.

I think the submission holds promise, but comprehensive editing is required.

Author Response

The authors should ask for the help of a native English-speaking proofreader because there are some linguistic mistakes that should be fixed.  The Title needs further thought - shortened and more accurate. The Abstract in its sub-sections needs re-organization and it does not adequately summarize the gist of the study. Also, the writing style of the manuscript is not overall  academic and formal. The article is proposed to be supplemented with a flowchart illustrating the research technique. A review of the literature is insufficient. It is critical to include some recent work (2018–2020) in the literature review. A literature review should be added in order to illustrate the central topic in a more detailed way. Some further explanations and interpretations are required for the results. It is recommended to include a well-organized discussion of the findings, strengths, and limitations of the present project with additional explanation/details and a conclusion with future work.  I think the submission holds promise, but comprehensive editing is required.

Authors' response: we thank the reviewer for the comment. The manuscript was now revised by a native English speaker. We remodeled the Abstract as suggested. We added a flowchart of the study (Figure 1). We made a complete review of the literature, and we summarized it in the Introduction section to illustrate the current knowledge on the impact of COVID-19 on adolescents and people with neuropsychiatric conditions. We completely remodeled the result section to present the data more clearly. We also added a new statistical analysis. We completely remodeled the Discussion section, also considering new findings and critically reviewing the current literature. We added strengths, limitations, and future prospective, as required.

Round 2

Reviewer 1 Report

I appreciate the authors' careful attention to suggestions.  I think this revised paper is more clear and methodologically sound.  Great work!

Author Response

We thank the reviewer for the comment.

Reviewer 2 Report

The authors have integrated many of the requested changes. 

I suggest its publication adding also a flowchart of the research procedure.

Author Response

We thank the reviewer for the suggestion. We added a flowchart of the research procedure in Figure 1.